# Exposure to Crude Oil-Related Volatile Organic Compounds Associated with Lung Function Decline in a Longitudinal Panel of Children

**DOI:** 10.3390/ijerph192315599

**Published:** 2022-11-24

**Authors:** Su Ryeon Noh, Jung-Ah Kim, Hae-Kwan Cheong, Mina Ha, Young-Koo Jee, Myung-Sook Park, Kyung-Hwa Choi, Ho Kim, Sung-Il Cho, Kyungho Choi, Domyung Paek

**Affiliations:** 1Department of Public Health and Environment, Kosin University, 194 Wachi-ro, Yeongdo-gu, Busan 49104, Republic of Korea; 2Department of Spatial & Environmental Planning, Chungnam Institute, 73-26 Yeonsuwon-gil, Gongju-si 32589, Republic of Korea; 3Department of Social and Preventive Medicine, Sungkyunkwan University School of Medicine, 2066 Seobu-ro, Jangan-gu, Suwon-si 16419, Republic of Korea; 4Department of Preventive Medicine, Dankook University College of Medicine, 119 Dandae-ro, Dongnam-gu, Cheonan-si 31116, Republic of Korea; 5Department of Internal Medicine, Dankook University College of Medicine, 119 Dandae-ro, Dongnam-gu, Cheonan-si 31116, Republic of Korea; 6Institute of Environmental Medicine, Seoul National University Medical Research Center, 103 Daehak-ro, Jongno-gu, Seoul 03080, Republic of Korea; 7Department of Public Health Sciences, Graduate School of Public Health, Seoul National University, 1 Gwanak-ro, Gwanak-gu, Seoul 08826, Republic of Korea; 8Institute of Health and Environment, Seoul National University, 1 Gwanak-ro, Gwanak-gu, Seoul 08826, Republic of Korea; 9Department of Environmental Health Sciences, Graduate School of Public Health, Seoul National University, 1 Gwanak-ro, Gwanak-gu, Seoul 08826, Republic of Korea; 10Division of Cancer Prevention, National Cancer Center, 323 Ilsan-ro, Ilsandong-gu, Goyang-si 10408, Republic of Korea

**Keywords:** benzene, cohort, ethylbenzene, health effect, long-term, pulmonary, students, toluene, xylene

## Abstract

Background: Children in the affected area were exposed to large amounts of volatile organic compounds (VOCs) from the Hebei Spirit oil spill accident. Objectives: We investigated the lung function loss from the exposure to VOCs in a longitudinal panel of 224 children 1, 3, and 5 years after the VOC exposure event. Methods: Atmospheric estimated concentration of total VOCs (TVOCs), benzene, toluene, ethylbenzene, and xylene for 4 days immediately after the accident were calculated for each village (n = 83) using a modeling technique. Forced expiratory volume in 1 s (FEV_1_) as an indicator of airway status was measured 1, 3, and 5 years after the exposure in 224 children 4~9 years of age at the exposure to the oil spill. Multiple linear regression and linear mixed models were used to evaluate the associations, with adjustment for smoking and second-hand smoke at home. Results: Among the TVOCs (geometric mean: 1319.5 mg/m^3^·4 d), xylene (9.4), toluene (8.5), ethylbenzene (5.2), and benzene (2.0) were dominant in the order of air concentration level. In 224 children, percent predicted FEV_1_ (ppFEV_1_), adjusted for smoking and second-hand smoke at home, was 100.7% after 1 year, 96.2% after 3 years, and 94.6% after 5 years, and the loss over the period was significant (*p* < 0.0001). After 1 and 3 years, TVOCs, xylene, toluene, and ethylbenzene were significantly associated with ppFEV_1_. After 5 years, the associations were not significant. Throughout the 5 years’ repeated measurements in the panel, TVOCs, xylene, toluene, and ethylbenzene were significantly associated with ppFEV_1_. Conclusions: Exposure to VOCs from the oil spill resulted in lung function loss among children, which remained significant up to 5 years after the exposure.

## 1. Introduction

On 7 December 2007, the oil tanker ‘Hebei Spirit’ crashed, spilling a total of 10,900 tons of crude oil only 10 km off the west shore of the Republic of Korea. This represented one-sixth of the Prestige oil spill in Spain, in which 67,000 tons were spilled [1], and one-sixtieth of the Deepwater Horizon oil spill in the US, in which 680,000 tons were spilled [2]. The volatile organic compounds (VOCs), which constituted a large portion of the spilled crude oil [3,4], were estimated to have completely evaporated within 4 days of the accident [5]. Among these VOCs, benzene had completely evaporated within 10 h, and toluene, ethylbenzene, and xylene had completely evaporated within 48 h. The other VOCs had fully evaporated within 4 days after the spill. The evaporated VOCs rapidly moved toward land on a southeast wind until 10 December 2007, and brought abnormally high concentrations of VOCs into several villages during the initial period after the spill. The concentrations of VOCs were 100-fold to 10,000-fold higher in these villages than in surrounding villages that had escaped the wind [5].

VOCs, compounds found in crude oil, are particularly toxic and pose a potential risk to human health [6]. One VOC, benzene, has been classified by the International Agency for Research on Cancer (IARC) as a human carcinogen (group 1). Toluene and ethylbenzene have been classified by the IARC as possible human carcinogens (group 2B). These VOCs have been shown to cause respiratory, hepatic, renal, neurologic, hematologic, reproductive, and developmental toxicity [7,8,9,10,11].

Considering the toxicity of VOCs on various organs in the human body, human inhalation of large amounts of VOCs evaporated by oil spills could not only lead to loss of lung function but also have long-term effects. Our previous study showed that children who lived within 2 km of the oil-contaminated coastline had a significant loss of lung function compared to those who lived more than 2 km 1 year after the oil spill [12]. The exposure to VOCs can be ubiquitous in residential indoor, ambient, and occupational environments. Building, furnishing, automobile exhausts, cigarette smoke, evaporative loss of fuels, and combustion processes in industries are the major sources of VOCs. There are some reports that exposure to VOCs can lead to deterioration of lung function in residential [13,14,15,16] or occupational environments [17]. In several studies showing the adverse effects of VOCs on lung function, we wondered whether the deterioration of lung function persisted over time.

To our knowledge, no health studies have looked into how long an accidental single exposure to large amounts of VOCs can affect lung function deterioration. Considering that even low levels of VOCs cause a loss of lung function in the general population [13,16], exposure to very high environmental concentrations of VOCs, albeit acutely, may have long-term adverse effects on lung function decline [18]. In addition, to the best of our knowledge, there are no children’s studies of a relationship between VOC exposure and lung function. Compared to adults, children are particularly vulnerable to health risks from exposure to excessive VOCs because they breathe in more air per unit of body mass and because their bodies detoxify many chemicals less efficiently [19]. Therefore, we investigated the long-term adverse effects on lung function at 1, 3, and 5 years after exposure to very high environmental concentrations of VOCs from the Hebei Spirit oil spill in a panel of 224 children who participated in all three surveys of all school-based students in the study area.

## 2. Materials and Methods

### 2.1. Study Population

The study area consisted of 83 small villages, and there were 13 elementary and 4 middle schools (Figure 1). The nearest village to the accident site at sea was 10 km away, and the farthest village was within 30 km [20]. The study area had a small population of children, which is a typical feature of rural villages, and that number declined in the aftermath of the oil spill. The present study on children is a part of Health Effect Research on Hebei Spirit Oil Spill (HEROS) study, of which design was described elsewhere [21]. Briefly, we administered school-based health surveys to all elementary and middle school students in the study area 1 year (June–November 2009), 3 years (March–April 2011), and 5 years (October–November 2013) after the oil spill. The participation rates were 87.7% (n = 828), 90.4% (n = 760), and 95.6% (n = 783), respectively, as calculated by comparing the number of participants to the population census numbers. A more detailed description can be found in our previous study [20]. Finally, the study subjects were 224 children who had undertaken lung function tests in all three surveys.

Each round of data collection was approved by the Institutional Review Board of Dankook University Hospital, Cheonan, Republic of Korea (IRB No. 2009-04-027, 25 May 2009; IRB No. 2011-03-081, 16 March 2011; IRB No. 2013-09-009, 14 October 2013). Each time, written informed consent was obtained from the participants or their parents.

### 2.2. Estimation of Exposure to VOCs

The exposure of the study subjects to oil-derived VOCs was estimated with a modeling technique integrating the weathering model and the California Puff (CALPUFF) air dispersion model; the methods have been described in detail in our previous study [5]. From these modeling processes, the ambient 1-h concentrations (µg/m^3^) in each village (total villages in the study area, n = 83) were calculated at a series of timepoints from the time of the accident. Subsequently, all hourly concentrations in each village from 7 December to 10 December 2007 (mg/m^3^·4 d) were added together to predict total VOC exposure levels. For children who were not yet in school at the time of the exposure, only the concentrations of VOCs at home were considered. For schoolchildren, the concentrations at home and at school were both considered, assuming the children spent 16 h a day at home and 8 h a day at school. Detailed VOC exposure assessment can be found in our previous asthma study [20], which assessed BTEX (benzene, toluene, ethylbenzene, and xylene) exposure in the same way as this study.

Total VOCs (TVOCs) included benzene, toluene, ethylbenzene, xylene, paraffin (C6–C12), cycloparaffin (C6–C12), and aromatic mono- and dicyclic components (C6–C11), as described in our previous study [5]. Finally, in this study, TVOCs, benzene, toluene, ethylbenzene, and xylene were examined as representatives of oil-derived VOCs.

### 2.3. Measurement of Lung Function

Spirometric measurements were performed according to the American Thoracic Society (ATS) guidelines [22]. A Microspiro HI-298 (Chest Corporation, Tokyo, Japan) was used in the first survey (after 1 year) and a Spirovit SP-1 (Schiller AG, Baar, Switzerland) was used in the second survey (after 3 years) and in the third survey (after 5 years) to measure forced expiratory volume in one second (FEV_1_). We calculated each participant’s percent predicted FEV_1_ (ppFEV_1_) on the basis of his or her age, height, and sex using the Global Lung Function 2012 equations, which are endorsed by the ATS and the European Respiratory Society (ERS) [23].

### 2.4. Statistical Analyses

Statistical analyses were performed with SAS version 9.4 (SAS Institute Inc., Cary, NC, USA). General characteristics were analyzed using the PROC MEANS and PROC FREQ procedures. All estimated VOC concentrations were log-transformed to approximate a normal distribution, and geometric mean (GM) and geometric standard deviation (GSD) were calculated. The cross-sectional relationship between exposure to VOCs and loss of lung function at 1, 3, and 5 years after the exposure, was analyzed by multiple linear regression (the PROC GLM procedure). A longitudinal relationship between exposure to VOCs and loss of lung function for the entire period of 5 years after the exposure, was analyzed by a linear mixed model (PROC MIXED procedure). An unstructured covariance matrix, which allows unequal variances over time and unequal covariances for each time combination, was assumed based on the smallest AIC. The estimated concentrations of VOCs were used as a continuous variable. Information concerning current smoking and second-hand smoke at home status was obtained from each subject through the survey. All analyses were adjusted for current smoking and second-hand smoke at home status.

## 3. Results

Study subjects (n = 224) were followed up for a mean period of 4.4 years from the first survey (after 1 year) to the third survey (after 5 years) (Table 1). Girls comprised 54.0% of the study subjects. The age range of the 224 children was from 4 to 9 years of age on the day of the accident (the time of very high environmental exposure to VOCs), from 6 to 11 years of age for the first survey (after 1 year), from 8 to 13 years of age for the second survey (after 3 years), and from 10 to 15 years for the third survey (after 5 years). A few children were current smokers: 0.9% of the study subjects as of the first survey, 0.5% as of the second survey, and 3.6% as of the third survey. A little less than half were second-hand smokers from family members who smoke in the home: 43.8% of the study subjects as of the first survey, 43.8% as of the second survey, and 48.2% as of the third survey. The average distance between the accident site and the house was 20.1 km, with a minimum of 12.1 km and a maximum of 35.8 km; the school was 20.0 km, with a minimum of 13.1 km and a maximum of 26.4 km. Only one out of 224 (0.4%) participated in clean-up activities to remove crude oil from the oil-contaminated coastline during the week following the accident, and was not considered as a covariate in the subsequent model.

The height, weight, and FEV_1_ of the subjects gradually increased during the follow-up period (Table 1). The mean FEV_1_ values on the first, the second, and the third survey were 1.69 L, 2.01 L, and 2.65 L, respectively. However, the mean ppFEV_1_ was significantly reduced over time (Table 1). The mean ppFEV_1_ was 96.5% as of the first survey, 92.0% as of the second survey, and 90.6% as of the third survey. After adjustment for smoking and second-hand smoke at home (Figure 2), the mean ppFEV_1_ of 224 subjects was 1 year (100.7%), 3 years (96.2%), and 5 years (94.6%) after theexposure to VOCs (*p* < 0.0001).

The cumulative estimated GM concentrations of atmospheric TVOCs, benzene, toluene, ethylbenzene, and xylene were 1319.5 mg/m^3^, 2.0 mg/m^3^, 8.5 mg/m^3^, 5.2 mg/m^3^, and 9.4 mg/m^3^ for 4 days immediately after the accident when very high environmental concentrations of VOCs were evaporated into the atmosphere, respectively (Table 2). Comparing the exposure magnitudes of individual VOC substances among the airborne TVOCs in 224 children, xylene was the highest, followed by toluene, ethylbenzene, and benzene in that order.

In a cross-sectional analysis to assess associations between exposure to VOCs and loss of lung function 1, 3, and 5 years after the very high environmental exposure to VOCs in the panel of 224 children (Table 3), the cumulative estimated concentrations of TVOCs, toluene, ethylbenzene, and xylene were significantly associated with the ppFEV_1_ after 1 year and after 3 years in smoking and second-hand smoke at home-adjusted models. The cumulative estimated concentrations of all VOC substances were not associated with the ppFEV_1_ after 5 years.

In a longitudinal analysis to examine associations between exposure to VOCs and loss of lung function up to 5 years after the very high environmental exposure to VOCs in the panel of 224 children (Table 4), the cumulative estimated concentrations of TVOCs, toluene, ethylbenzene, and xylene were significantly associated with the ppFEV_1_ after adjustment for smoking and second-hand smoke at home.

To examine whether there are age differences in the association between exposure to VOCs and lung function decline, stratified analysis was performed based on the age at the time of the accident. There were no differences both in the cross-sectional analysis (Appendix A) and in the longitudinal analysis (Appendix A).

## 4. Discussion

Exposure to crude oil-related VOCs caused loss of lung function in the longitudinal panel of 224 children up to 5 years after the one-time exposure to large amounts of VOCs. The lung function, in the longitudinal panel of 224 children, deteriorated and did not recover during the 4-year follow-up period. Cross-sectional associations between exposure to VOCs, such as TVOCs, toluene, ethylbenzene, and xylene, and loss of lung function were significant at 1 and 3 years after the one-time very high environmental exposure to VOCs; they were not significant at 5 years. Longitudinal associations between exposure to VOCs, such as TVOCs, toluene, ethylbenzene, and xylene, and loss of lung function were statistically significant in the panel of 224 children.

It is thought that the high-dose exposure acted as a decisive factor in the association between exposure to VOCs and lung function decline even after 5 years, even if it was a one-time event. VOCs are main components of crude oil [24]. VOCs, such as benzene, toluene, and xylene, are lower-molecular-weight aromatic hydrocarbons. VOCs evaporate within hours after the oil reaches the shore [25] and therefore are emitted only from a narrow area [26]. It is estimated that the VOCs from the Hebei Spirit oil spill polluted several villages within hours or up to a few days [5]. In the case of the Deepwater Horizon oil spill, all VOCs appeared to evaporate before reaching the shore [27]. This difference is due to the proximity of the residential areas to the accident point. The location of the Hebei Spirit oil spill accident was just 5 miles off the coast, while the location of the Deepwater Horizon oil spill was 50 miles off the coast [2] and the Prestige oil spill was 130 miles off the coast [1]. Consequently, it was estimated that children living in the vicinity of the accident site were exposed to very high environmental levels of VOCs through inhalation within hours or up to a few days after the Hebei Spirit oil spill accident [5]. Due to high-dose exposure to VOCs, albeit a single exposure, decreased lung function appears to have been observed up to 5 years in this study.

For each VOC substance, the exposure response relationship was apparent in reducing lung function. The present study shows that benzene had less of an effect than the other VOCs, such as toluene, ethylbenzene, and xylene, on the loss of lung function in children. This difference is probably due to the relatively smaller amount of benzene exposure. The exposure to benzene was smaller than any of the other VOCs at home and at school during the initial oil-spill period, because the benzene completely evaporated within 10 h, while the oil and volatiles did not reach the shore of the residential area until 14 h after the spill. Consequently, the total amount of benzene exposure in children (mean: 2.3 mg/m^3^·4 d) was one-fifth that of xylene (11.2 mg/m^3^·4 d). For this reason, although benzene is a highly toxic substance designated as a Class 1 carcinogen by the IARC, WHO, it appears that it was not associated with a decrease in lung function.

The time elapsed after exposure to VOCs also appears to be an important factor in the association with the loss of lung function. From the cross-sectional analysis, the associations between exposure to VOCs and the loss of lung functions were statistically significant at 1 and 3 years after the exposure, but were not statistically significant at 5 years after the exposure. Nevertheless, children’s lung function was not recovered even 5 years after the exposure. The ppFEV_1_ after adjustment for smoking and second-hand smoke was 100.7%, 96.2%, and 94.6% after 1 year, 3 years, and 5 years, respectively (*p* < 0.0001).

Estimating the human exposure route that resulted in the deterioration of lung function due to the toxic components in the oil from the accident, it seems that it is due to the exposure of a large amount of VOCs in the early stage of the accident. The VOCs completely evaporated in the initial period of the oil spill, within a few hours or days. There are main exposure routes, such as participation in clean-up activity for a long and initial high exposure from air via inhalation resulting from living close to the oil-spill site. Participation in oil-spill clean-up work is a well-known major risk factor in adults [28]; however, the children in this survey hardly participated in clean-up work (0.4%). Thus, initial very high environmental exposure to VOCs via inhalation over a few days immediately after the spill seems to result in the loss of lung function in children.

Our study used FEV_1_ and ppFEV_1_ as indicators of lung function. If the exposure route of VOCs is assumed to be inhalation, FEV_1_, which looks at the effects on major airways, was considered the most appropriate indicator rather than forced vital capacity (FVC), which looks at the effects on pulmonary parenchyma. Yoon et al. also showed that FVC did not associate with VOC exposure in contrast to FEV_1_ [14]. In our previous study of 1 year after the Hebei Spirit Oil spill, among the measured parameters of lung function, such as FVC, percent predicted FVC (ppFVC), FEV_1_, ppFEV_1_, and FEV_1_/FVC ratio, only the ppFEV_1_ parameter was significantly associated with the oil exposure in children [12]. Aldrich et al. also used the FEV_1_ parameter to study lung function in rescue workers at the World Trade Center after 7 years [29].

Some studies have reported a relationship between VOC exposure and a decline in lung function, thereby supporting our results. Our previous results showed significantly lower lung function in children who lived <2 km from the oil-contaminated coastline than in those who lived ≥2 km from the oil-contaminated coastline 1 year after the Hebei Spirit oil spill, which is in line with the present study [12]. In the occupational setting, a significant reduction in FEV_1_ is observed with increased urinary VOC metabolites among the petrol pump workers; some lung function parameters showed gradual reduction as the duration of exposure increased [17]. Exposure to traffic-related VOCs was associated with a decrease in FEV_1_ when monitored before cycling and 1–4 h after starting cycling, demonstrating that VOCs have acute health effects on lung function [15]. In a cross-sectional study designed to represent the Canadian population, exposure to residential VOCs, ubiquitous in homes, negatively influenced lung function [16]. In a longitudinal panel study of 154 elderly people, metabolites of toluene and xylene were significantly associated with a decrease in FEV_1_ [14]; it means that even ordinary low-exposure to VOCs can affect the deterioration of lung function in the general population. Additionally, in the U.S. population, blood concentrations of 1,4-dichlorobenzene were significantly associated with reduced FEV_1_ [13].

A better understanding of the long-term effects of a single exposure to high-dose VOCs on the deterioration of lung function in our study can be achieved by referring to the study of previous disasters. Referring to studies related to the Prestige oil spill accident in 2002, persistent respiratory health effects were reported 5 years after the Prestige oil spill in adults [30]; these effects disappeared 6 years after the spill [31]. One notable disaster, the terrorist attacks on the World Trade Center on 11 September 2001, in the USA, brought a large decline in lung function in rescue workers at the site during the first year; these declines were persistent and did not reverse over the next 6 years [29]. These studies, including our results, show that even a single exposure can have long-lasting adverse health effects when exposed to large amounts of toxic substances.

The main mechanism of VOC toxicity on lung function could be exacerbated by oxidative stress [32]. The study, which showed not only the association between exposure to VOCs and increased systemic level of oxidative stress but also the association between markers of oxidative stress and parameters of lung function supports this hypothesis [14]. As VOCs are gaseous pollutants, their primary target tissue is the lung, and breathing elevated concentrations of VOCs could have direct toxic effects on cells through some possible mechanisms, such as inflammation and DNA repair pathways, in addition to oxidative stress [18,33]. In addition to genetic toxicity, epigenetic mechanisms appear to be involved. A recent epigenetic study showed that higher methylation levels in promoters of genes involved in cellular proliferation/DNA repair (TOP2A), oxidative stress (SOD1), and inflammation (TNF-α) in leukocytes from people occupationally exposed to VOCs [34].

A particular strength of this study was the longitudinal panel design that followed the children over a four-year period. To the best of our knowledge, this study is the first to observe long-term lung health consequences after one-time exposure to large amounts of oil-derived VOCs in a longitudinal panel of children. There are some studies examining the health effects of exposure to VOCs, but it seems that most of them were short-term studies or that the subjects were not children. Children may face a greater health risk than adults from an identical amount of exposure to VOCs because they breathe in more air per unit of body mass and because their bodies detoxify VOCs less efficiently [19]. Our study results came from repeated measures across a total of three time points to assess long-term lung health in children, despite the difficulty of enrolment and tracking of a cohort for an environmental epidemiology study, e.g., the 2010 Deepwater Horizon oil spill in the Gulf of Mexico off the coast of the USA [35]. In addition, this study is considered to have no selection bias, since all children in the exposed area were investigated through the entire school-based surveys. Nevertheless, there are some limitations. The levels of VOC exposure in the 4 days immediately after the spill were estimated from a modeling technique because actual ambient exposure assessments were not obtained in the initial period. In addition, lung function data from before or immediately after the oil spill were not available. There is also a limitation that all study subjects were in the exposed group and could not be compared with the non-exposed control group.

In conclusion, completely evaporated VOCs in the initial period after an oil spill can reduce lung function in children for at least 5 years. The airborne exposure levels of VOCs in the first few days after an oil spill are generally abnormally high. Therefore, children, a group vulnerable to environmental toxicant exposure, could endure adverse lung health impacts of oil spills, even without direct contact with oil chemicals, such as from clean-up activity, and these effects could persist over the next several years. Guidelines for preventive management and the continued surveillance of lung function in children are required for this spill. Further research related to BTEX exposure from oil spills or BTEX exposure generally is necessary and will aid in the assessment and treatment of those exposed to past and future spills.

## Figures and Tables

**Figure 1 ijerph-19-15599-f001:**
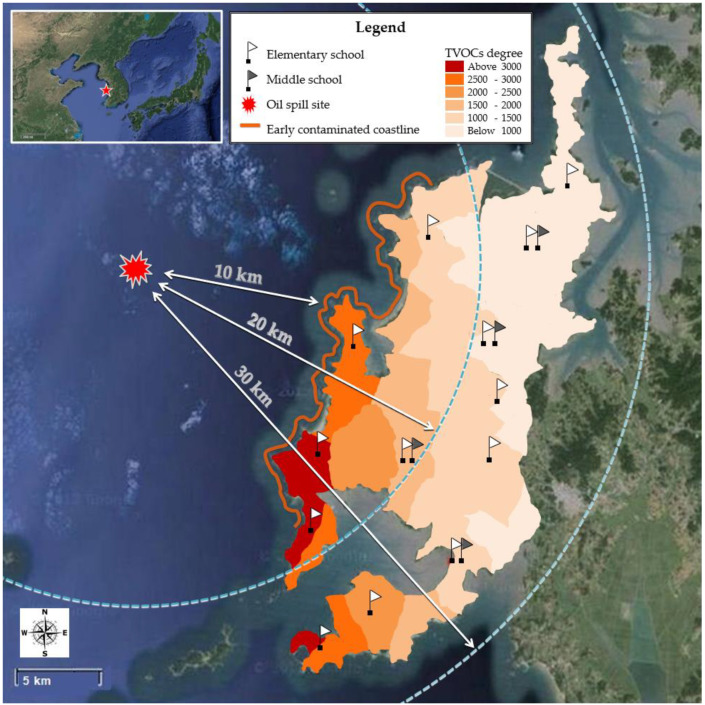
Distribution of TVOCs estimated concentration in the study area.

**Figure 2 ijerph-19-15599-f002:**
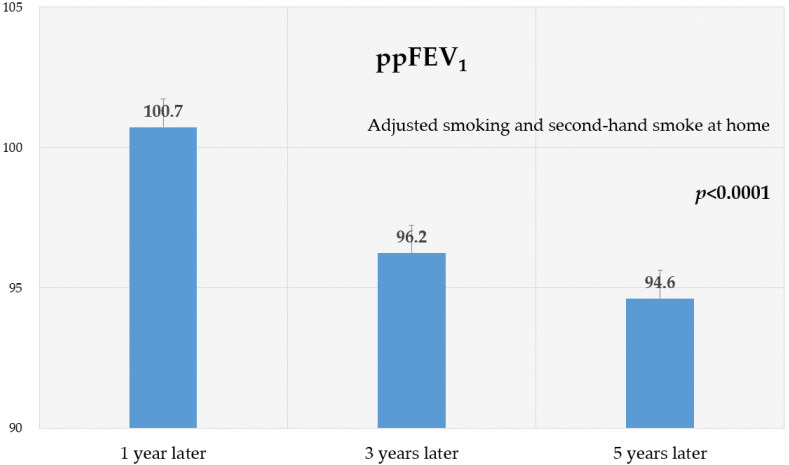
Lung function in children over time (n = 224). ppFEV_1_: percent predicted FEV_1_. Linear mixed models after adjusting for smoking (no, yes), second-hand smoke at home (no, yes), and visit time (first survey, second survey, third survey) were used with an unstructured covariance matrix.

**Table 1 ijerph-19-15599-t001:** Characteristics of the participants including lung function.

Characteristic	Total (n = 224)
Girls	121 (54.0%)
Age (years)	
Exposure to VOCs ^a^	7.1 (4−9)
1 year later ^b^	8.7 (6−11)
3 years later ^c^	10.5 (8−13)
5 years later ^d^	13.0 (10−15)
Height (cm)	
1 year later ^b^	134.3 ± 9.4
3 years later ^c^	145.2 ± 9.9
5 years later ^d^	159.1 ± 8.2
Weight (kg)	
1 year later ^b^	32.4 ± 9.7
3 years later ^c^	40.3 ± 12.4
5 years later ^d^	53.7 ± 14.3
Current smokers	
1 year later ^b^	2 (0.9%)
3 years later ^c^	1 (0.5%)
5 years later ^d^	8 (3.6%)
Second-hand smoke at home	
1 year later	98 (43.8%)
3 years later	98 (43.8%)
5 years later	108 (48.2%)
Distance from the exposure to VOCs	
Home	20.1 (12.1−35.8)
School	20.0 (13.1−26.4)
FEV_1_ (L)	
1 year later ^b^	1.69 ± 0.36
3 years later ^c^	2.01 ± 0.50
5 years later ^d^	2.65 ± 0.62
ppFEV_1_ (%)	
1 year later ^b^	96.5 ± 10.6
3 years later ^c^	92.0 ± 14.5
5 years later ^d^	90.6 ± 12.9

ppFEV_1_: percent predicted FEV_1_. Data are presented as n (%), mean (min−max), or mean ± SD. ^a^ Immediately after the accident, 2007; ^b^ first survey, 2009; ^c^ second survey, 2011; ^d^ third survey, 2013.

**Table 2 ijerph-19-15599-t002:** Cumulative estimated concentrations of atmospheric total volatile organic compounds (TVOCs), benzene, toluene, ethylbenzene, and xylene over the first 4 days after the exposure to VOCs from the Hebei Spirit oil spill.

Cumulative Estimated Concentration (mg/m^3^·4 d)	GM (GSD)	Median (Q1–Q3)	Min−Max
TVOCs ^a^	1319.5 (1.8)	1157.0 (924.3–2525.5)	466.7–3446.3
Benzene	2.0 (1.6)	1.8 (1.5–2.6)	0.9–9.5
Toluene	8.5 (1.8)	6.9 (5.9–13.9)	3.0–23.0
Ethylbenzene	5.2 (1.8)	4.6 (3.6–10.0)	1.8–13.6
Xylene	9.4 (1.8)	8.2 (6.5–18.5)	3.3–24.8

GM: geometric mean, GSD: geometric standard deviation, Q1: 25th percentile, Q3: 75th percentile. ^a^ TVOCs: total volatile organic compounds including benzene, toluene, ethylbenzene, xylene, paraffin (C6−C12), cycloparaffin (C6−C12), and aromatic mono- and dicyclic components (C6−C11).

**Table 3 ijerph-19-15599-t003:** Cross-sectional associations between exposure to VOCs and loss of lung function 1 year, 3 years, and 5 years after the very high environmental exposure to VOCs from the Hebei Spirit oil spill (n = 224).

	ppFEV_1_					
	1 Year Later ^a^		3 Years Later ^b^		5 Years Later ^c^	
Cumulative Estimated Concentration (mg/m^3^·4 d) ^d^	β (SE)	*p*-Value	β (SE)	*p*-Value	β (SE)	*p*-Value
TVOCs ^e^	−3.8 (1.2)	0.002	−5.4 (1.7)	0.001	−2.4 (1.5)	0.11
Benzene	−2.5 (1.5)	0.09	−3.5 (2.0)	0.08	−1.5 (1.8)	0.39
Toluene	−3.6 (1.2)	0.003	−5.1 (1.7)	0.003	−2.2 (1.5)	0.14
Ethylbenzene	−3.8 (1.2)	0.002	−5.3 (1.6)	0.001	−2.4 (1.5)	0.11
Xylene	−3.8 (1.2)	0.002	−5.2 (1.6)	0.002	−2.4 (1.5)	0.11

ppFEV_1_: percent predicted FEV_1_. Log scale of TVOCs, benzene, toluene, ethylbenzene, and xylene were used as independent variables. Estimates and *p*-Values were from generalized linear regression analysis adjusted for current smoking (no, yes) and second-hand smoke at home (no, yes). ^a^ First survey, 2009; ^b^ Second survey, 2011; ^c^ Third survey, 2013. ^d^ Cumulative estimated concentrations over the first four days after the oil spill were used. ^e^ TVOCs: total volatile organic compounds including benzene, toluene, ethylbenzene, xylene, paraffin (C6−C12), cycloparaffin (C6−C12), and aromatic mono- and dicyclic components (C6−C11).

**Table 4 ijerph-19-15599-t004:** Longitudinal associations between exposure to VOCs and loss of lung function up to 5 years after the very high environmental exposure to VOCs from the Hebei Spirit oil spill (n = 224).

	ppFEV_1_	
Cumulative Estimated Concentration (mg/m^3^·4 d) ^a^	β (SE)	*p*-Value
TVOCs ^b^	−3.7 (1.1)	0.0009
Benzene	−2.4 (1.3)	0.07
Toluene	−3.5 (1.1)	0.002
Ethylbenzene	−3.7 (1.1)	0.0008
Xylene	−3.3 (1.1)	0.003

ppFEV_1_: percent predicted FEV_1_. Log scale of TVOCs, benzene, toluene, ethylbenzene, and xylene were used as independent variables. Linear mixed models after adjusting for smoking (no, yes), second-hand smoke at home (no, yes), and visit time (first survey, second survey, third survey) were used with an unstructured covariance matrix. ^a^ Cumulative estimated concentrations over the first four days after the oil spill were used. ^b^ TVOCs: total volatile organic compounds including benzene, toluene, ethylbenzene, xylene, paraffin (C6−C12), cycloparaffin (C6−C12), and aromatic mono- and dicyclic components (C6−C11).

## Data Availability

Associated data are provided in the manuscript.

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
