# Peer review of "Exposure to Crude Oil-Related Volatile Organic Compounds Associated with Lung Function Decline in a Longitudinal Panel of Children"

_ijerph, 2022, doi:10.3390/ijerph192315599_

Round 1
Reviewer 1 Report
This paper focuses on the effect of atmospheric levels of several volatile organic compounds (VOCs) on the lung function of 224 children who live in the area affected by Hebei Spirit oil spill accident. Therefore, this study assesses the impact on respiratory health of exposure to a severe environmental pollution event during childhood. Although, the analysis of volatile organic compounds in the exhaled air of the children in the study would have been very enriching for the paper. The article is concise, understandable, well written and will be of interest. However, I have a few comments:
1. Although the models used in this study are adjusted for smoking, the influence of environmental tobacco smoke has not been considered. Second-hand smoke may also decrease lung function.
2. Have the children developed any respiratory diseases and/or allergies during the years following the accident? This information could complement the findings provided by lung function screening.
3. The relationship between lung function, inflammation and oxidative stress needs to be described in more detail (10.1016/j.envpol.2021.117215 and 10.1002/ppul.25849)
4. Did the age of the children at the time of the accident influence the degree of lung function decline?
Reviewer 2 Report
I have attached a word file with my critiques and comments to the authors. Please let me know if there are any issues with the file or you would like the file as a PDF.

Reviewer 3 Report
I find results very interesting and worth to be published, however, I have some comments and recommendations; they are related with the fact that authors could actually atribute the changes in respiratory evaluations to the unique exposure from the mentiones accident. Firstable, we do not know if participants were exposed even yo low levels of VOCs in their place of residence (as part of enviromental pollution or pollution from motor vehicles, even ,motor operated boats) after de accident, for instance, they need to give details about the place of residence (mostly rural/urban, if kids help their parents in labor activities, etc). I think a limitation of the study is that you do not included a control group; then you need also to give us more informtion regarding if the ppFV1 values observed are in a range condired like an "affection" compared to values in other children. Again: hoe can we be sure that modifications are due only to the exposure for the accident? Do we have knowledge that that reduction is due to a "normal, inevitable" reduction in chhildren from that ages you need to elaborate about this. For instance: in Zock et al (2014), when they report FEV1/FVC ratio (%) values (Figure 2) we can see that changes are similar in both the exposed group but also in the non-exposed; actually, it is more evident in the non-exposed group. In that same line, I think you need stronger arguments for claiming that changes are due to the exposure in the accident day (also, stimated are not so high, when compared to some chronic exposures in occupationaly exposed adults)
Round 2
Reviewer 1 Report
No comments